# Improvement of the Stability and Optical Properties of CsPbBr_3_ QDs

**DOI:** 10.3390/nano13162372

**Published:** 2023-08-18

**Authors:** Jiaming Wang, Li Zou, Meili Yang, Jiajie Cheng, Yufan Jiang, Guangdong Huang, Jingjing Dong

**Affiliations:** School of Science, China University of Geosciences Beijing, No. 29 College Road, Haidian District, Beijing 100083, China

**Keywords:** perovskite quantum dots, CsPbBr_3_, PMMA, optical properties, stability

## Abstract

All-inorganic perovskite quantum dots (CsPbX_3_ QDs) (X = Cl, Br, I) have the advantages of adjustable emission position, narrow emission spectrum, high fluorescence quantum efficiency (PLQY), easy preparation, and elevated defect tolerance; therefore, they are widely used in optoelectronic devices, such as solar cells, light-emitting diodes, and lasers. However, their stability still constrains their development due to their intrinsic crystal structure, ionic exchange of surface ligands, and exceptional sensitivity to environmental factors, such as light, water, oxygen, and heat. Therefore, in this paper, we investigate the stability improvement of CsPbX_3_ QDs and apply fabricated high-efficiency, stable perovskite QDs to solar cells to improve the performance of the cells further. In this paper, we focus on CsPbBr_3_ QDs with intrinsic extreme stability and optimize CsPbBr_3_ QDs using strategies, such as Mn^+^ doping, ligand regulation, and polymer encapsulation, which can improve optical properties while ensuring their stability. The test results show that the above five methods can improve the strength and luminescence performance of QDs, with the best stability achieved when PMMA encapsulates QDs with a ratio of PMMA = 2:1 and PLQY increases from 60.2% to 90.1%.

## 1. Introduction

All-inorganic perovskite materials are being hailed as potential materials for next-generation light-emitting devices (LEDs), due to their high carrier mobility, high radiation recombination efficiency, high color purity, and parameter bandgap [1,2,3]. All-inorganic metal halide perovskite quantum dots (QDs) were first reported by Protesescu et al. in 2015 [4], and since then, inorganic perovskite materials have sparked interest among researchers due to their superior optical characteristics and affordable production. The all-inorganic metal halide perovskites (CsPbX_3_, X = Cl, Br, I), a novel kind of direct bandgap semiconductor material, provide certain distinctive characteristics, including excellent quantum efficiency, significant optical absorption coefficient, small emission peak half-width, and variable luminescence wavelength [5,6,7]. It has broad applications in light-emitting diodes, solar cells, and photodetectors [8,9,10,11]. Stoumpos et al. (2013) used the CsPbBr_3_ single crystal for high-energy radiation detection [12]. Wang et al. (2017) fabricated blue, green, and red VCSELs (vertical cavity surface emitting lasers) by using CsPb(Br/Cl)_3_, CsPbBr_3_, and CsPb(I/Br)_3_ as active materials, which exhibited low threshold, directional output, and good stability [13]. Yi-Xin Chen et al. (2020) also found that the CsPbBr_3_ halide perovskite photocatalyst significantly affected the CO_2_ reduction performance, achieving a higher reaction rate and long-term stability [14]. Liu et al. (2021) employed strained perovskite QDs as nucleation sites to increase the stability and uniformity of the film crystallization [15]. The red CsPbI_3_ QDs films-based LED demonstrated external quantum efficiency (EQE) of 18% and retained half of the initial brightness even after 2400 h of operation due to the method’s decreased gradient crystallization and suppression of halide segregation. All of these papers show that the perovskite semiconducting material is perfect for use in optoelectronic devices.

CsPbX_3_ QDs have low stability due to their high sensitivity to the outside environment, which limits the commercial use of perovskite [16]. To increase the strength of perovskite QDs, more research is still needed. As reported, the common routes for enhancing the stability of all-inorganic perovskite QDs include surface ligand modification, doping engineering, and surface coating. Zou et al. (2017) tried to substitute comparable ions for some of the elements at the A and B sites by ion doping [17]. The A-site doping aids in raising the tolerance factor, and further enhances the stability of perovskite material. The B-site doping primarily substitutes Pb^2+^ with smaller ion radii, changing the length of the B-X bond in the process, which increases phase stability and reduces the number of dangerous Pb^2+^ ions. Surface ligand modification aims to introduce ligands that firmly bind to the uncoordinated lead and halide ions on the QDs surfaces, passivating dangling bonds and defect sites, thereby reducing non-radiative channels and improving optical performance [18,19,20]. Liu et al. reported a surface coating method, which adopted transparent wide-bandgap materials to encapsulate and shield the perovskite QDs from the external environment [21]. This sharply reduces the degradation of QDs, thereby improving their water and oxygen resistance. Additionally, it prevents the loss of quantum efficiency brought on by aggregation and ion exchange effects. After modification, the photoluminescence quantum yield (PLQY) and stability of perovskite QDs are both significantly increased.

To improve the stability of CsPbBr_3_ PQDs, five methods were adopted to optimize the quantum dots: (1) Different amounts of Mn^2+^ doping; (2) different proportions of PEABr partially substituted ligand OAm before QDs synthesis; (3) different proportions of PEABr and OAm ligand exchange after QDs synthesis; (4) DA substituted ligand OA; (5) different proportions of PMMA encapsulation QDs. This paper focuses on the stability and fluorescence enhancement strategies of all-inorganic CsPbBr_3_ QDs, primarily describing the aspects of doping engineering, surface ligand modification, and surface coating strategies.

## 2. Materials and Methods

### 2.1. Materials

N, n-dimethylformamide (DMF, 99.8%), Polymethyl methacrylate (PMMA, 80%), Manganese bromide dihydrate (MnBr_2_·2H2O, 90%), Ethyl acetate (EA, 80%), Oleic acid (OA, 99%), and Oleamine (OAm, 80%) were purchased from Aladdin Corporation (Shanghai, China). Lead bromide (PbBr_2_, 99.9%), Cesium bromide (CsBr, 99.9%), 2-Phenethylammonium bromine (PEABr, 99.9%), and 2-hexyl capric acid (DA, 90%) were purchased from Xi’an Polymer Light Technology Crop. (Xi’an, China). Moreover, toluene was supplied by Sinopharm Group Chemical Reagent Beijing Co. LTD (Beijing, China). All of these materials were used without additional treatment or purification.

### 2.2. Method

#### 2.2.1. Preparation of Original CsPbBr_3_ PQDs

First, PbBr_2_ (0.2 mmol) and CsBr (0.2 mmol) were dissolved in DMF (5 mL) to obtain the precursor solution. Then, OAm (0.25 mL) and OA (0.5 mL) were added to the precursor solution and stirred until the solution became clear. Briefly, 1 mL of the above solution was taken and slowly injected into a 10 mL solution of toluene under intense stirring at 1200 rpm. After 5 s, the solution turned bright green and the solution of CsPbBr_3_ PQDs was obtained, as shown in Figure 1.

#### 2.2.2. Preparation of Mn^2+^ Doped CsPbBr_3_ PQDs

The precursor solution was prepared by dissolving 9 mg, 12 mg, and 15 mg of MnBr_2_·4H_2_O with PbBr_2_ (0.2 mmol) and CsBr (0.2 mmol) in DMF (5 mL), respectively. The additional preparation procedure remained the same.

#### 2.2.3. Preparation of CsPbBr_3_ PQDs Partially Substituted by PEABr Ligand OAm

First, the PEABr solution was prepared: 1.52 mg of PEABr powder and 50 μL of OA were dissolved in 4 mL ethyl acetate solution. Then, the transparent and clarified PEABr solution was obtained by ultrasound for 5 min. During the preparation of the precursor solution, OAm was replaced by PEABr solution with molar ratios of 25%, 50%, 75%, and 100%; all other conditions remained unchanged.

#### 2.2.4. Preparation of CsPbBr_3_ PQDs Exchanged by PEABr with OAm Ligand

After the synthesis of quantum dots, PEABr solution with molar ratios of 25%, 50%, 75%, and 100% to OAm was added for ligand exchange with OAm; all other conditions remained unchanged.

#### 2.2.5. Preparation of CsPbBr_3_ PQDs of DA Substituted Ligand OA

During the preparation of the precursor solution, OA was replaced by DA solution with an equal molar ratio; all other conditions remained unchanged.

#### 2.2.6. Preparation of PMMA Encapsulated CsPbBr_3_ PQDs

First, a PMMA solution was prepared: 50 mg of solid PMMA material was dissolved in 1 mL toluene solution, and then heated and stirred at constant temperature in an 80 °C oil bath until entirely dissolved. Upon cooling, a PMMA solution can be obtained. After QD synthesis, PMMA is added to the QD solution with the volume ratio of QD: PMMA = 1:2, 1:1, 2:1, and mixed uniformly to obtain solutions of CsPbBr_3_ PQDs encapsulated in different proportions.

#### 2.2.7. Characterization

The structure and components were tested by X-ray diffraction technique (XRD, D8 fox, Bruker AXS Inc., Fitchburg, WI, USA). The light absorbance spectra were determined by UV-vis absorption spectra (Cary 7000, Agilent Technologies, Santa Clara, CA, USA). Photoluminescence (PL) and time-resolved photoluminescence (TRPL) emission spectra were conducted by using Nano LOG-TCSPC with a 780 nm laser. The size and distribution of the quantum dots were observed using the Transmission electron microscope (TEM, FEI TALOS F200X, Thermo Fisher Scientific, Waltham, MA, USA).

## 3. Results and Discussion

### 3.1. Original CsPbBr_3_ PQDs

We examined the fluorescence intensity of the QD solution after 1 and 8 months, respectively, and compared it with the original solution to investigate the stability of CsPbBr_3_ PQDs in air, as shown in Figure 2 and Table 1. According to the experimental findings, CsPbBr_3_ PQDs grow and cluster more with time, while their fluorescence intensity diminishes due to their inherent instability. To determine the size of the pure CsPbBr_3_ PQDs, we also conducted TEM and HRTEM studies on them. Additionally, from Figure 2g,h, we can obtain that the QDs have a size of around 10 nm and a lattice spacing of 0.46 nm.

To explore the optical properties of CsPbBr_3_ PQDs, we tested the PL emission spectra of the original CsPbBr_3_ PQDs after the initial and 3 day epochs. Based on the data presented in Figure 3a and Table 2, it is evident that the luminescence intensity decreases significantly after 3 days. This is due to the non-radiative transition of the defect state. The fully inorganic halide-encapsulated crystalline quantum dots also experience a loss in brightness due to their instability in ambient conditions. Additionally, due to their weak internal ionic bonding and high surface energy, these quantum dots agglomerate and experience structural deterioration. The quantum dots’ capacity to absorb and emit energy declines, their surface energy rises, more of them are gathered together, and their emission spectra change toward lower energies or toward becoming red [22]. In Figure 3a, it can be observed that the photometric wavelength’s position has been redshifted from 525 nm to 534 nm. Additionally, the QD PLQY is only 60.2%.

In this paper, the XRD pattern of the CsPbBr_3_ PQDs that was initially tested is displayed in Figure 3b and compared to the standard CsPbBr3 card. The diffraction peaks at 26.5° and 37.8° correspond to the (111) and (211) crystal faces of perovskite [23], which suggests that the quantum dots acquired are indeed CsPbBr_3_ PQDs.

### 3.2. Mn^2+^ Doped CsPbBr_3_ PQDs

MnBr_2_ doping was used to reduce the lattice defects of QDs to increase the stability and luminous performance of CsPbBr_3_ PQDs. Part of the Pb^2+^ was replaced with Mn^2+^, which was effectively doped into the lattice of PQDs. The doping of Mn^2+^ resulted in the lattice contraction, which would decrease the size of quantum dots since the Mn^2+^ ionic radius is less than Pb^2+^. Additionally, the amount of uncoordinated Pb^2+^ on the surface of the PQDs has been significantly reduced as a result of the doping of Mn^2+^, and Mn^2+^ has a higher affinity for the halogen group elements. These factors increase the formation energy, reduce the density of defect states on the surface, and improve the stability of PQDs, all of which contribute to a higher PLQY. As shown in Figure 4 and Table 1, the Mn^2+^ doped quantum dot solution is clearer and more transparent under daylight and UV irradiation than those in Group I, and the color of the luminescence changes to a short-wave blue, showing that the volume of the doped quantum dots is reduced and the luminescence performance is enhanced. Over time, the doped QDs solution turns turbid and loses fluorescence intensity. Since the degree of decline is less than those in Group I, the doped QDs stability and luminous performance have improved. Over time, the doped QDs solution turns turbid and loses fluorescence intensity.

To examine how different amounts of Mn^2+^ doping affects the optical properties of quantum dots (QDs), we tested and compared the PL emission spectra of QDs doped with 9 mg, 12 mg, and 15 mg of MnBr_2_ before doping. Our results, as demonstrated in Figure 5a, indicate a significant enhancement in PL emission intensity after doping. Additionally, the emission wavelength shifted to blue, indicating a reduction in the quantum dot size and an improvement in luminescence performance. Notably, the most significant improvement occurred at a doping level of 9 mg, as evidenced in Table 2, where the PLQY increased from 60.2% pre-doping to 78%. The specific explanation is: B^−^ site doping will considerably affect the exciton band gap, namely, the partial substitution of smaller Mn^2+^ for Pb^2+^ causes lattice shrinkage, leading to the blue shift of the spectrum, namely, lattice shrinkage and electron coupling of the [MnBr_6_]^4−^ octahedral [PbBr_6_]^4−^ skeleton, and the doped Mn^2+^ will not generate new radiation centers or trap states. This helps to reduce the non-radiative convergence caused by surface defects and helps to improve PL performance and PLQY [24]. In addition, the introduction of Br^−^ enriches the surface of the QD with halogens, and these Br^−^ are connected to the cation Pb^2+^ and reduce the formation of halogen vacancies, thus passivating surface defects and enhancing radiative convergence. At the same time, the formation of PbBr_n_ compounds can also effectively act as a shell for the QD, forming a core-shell structure that blocks the invasion of external water and oxygen and improves stability.

The stability of doped quantum dots was tested, as seen in Figure 5b. After 3 days, there was no noticeable decrease in luminous intensity, and the luminous wavelength shifted from 517 nm to 525 nm. This was an improvement compared to the agglomeration phenomenon observed before doping. The non-radiative transition rate of doped defective states was effectively inhibited, contributing to improved stability. The optical properties and stability of the QDs were also improved. As can be seen from the absorption map in Figure 5c, the absorption of the doped QDs is significantly higher than the pre-doped case, which also indicates an improvement in the optical properties after doping, in agreement with previous results. From the XRD pattern of Figure 5d, it can be seen that the diffraction peaks at 26.5° and 37.8° correspond to the (111) and (211) crystal planes of perovskite, indicating that the crystal structure of doped quantum dots does not alter significantly.

### 3.3. PEABr Partially Replaces the CsPbBr_3_ PQDs of the Ligand

To decrease non-radiative channels and improve optical characteristics, surface ligand modification intends to introduce ligand passivation of dangling bonds and defective sites of uncoordinated lead and halogen ions that are tightly attached to the QD surface. For the tests in this study, the replacement rates of the ligand PEABr on OAm were set at 25%, 50%, 75%, and 100%. The findings are displayed in Figure 6 and Table 1. According to the experimental findings, the PQD solution following partial ligand replacement is clear green in natural light and vivid blue-green in ultraviolet light. The QDs solution when PEABr replaces the ligand is cleaner and more transparent than those in Group I, and the color of the luminescence changes to a short-wavelength blue, showing that the quantum dot size decreases and the luminescence performance increases. Even though the ligand was only partially substituted, the PQDs solution eventually begins to aggregate and loses fluorescence intensity, but to a lesser extent than those in Group I. This suggests that following ligand replacement, the stability and luminescence characteristics of the quantum dots were enhanced.

As shown in Figure 7a, the PL emission intensity increases significantly after 25% and 50% substitution, especially when 50% substitution is applied, and the emission wavelength is blue-shifted, indicating a reduction in the QD size and an improvement in the luminosity performance, as shown in Table 2. At this time, PLQY increased from 60.2% before replacement to 85.2%. However, when the substitution ratio is increased to 75 percent, the emission wavelength is redshifted and the PL emission intensity is reduced, and even a fluorescence quench occurs when the substitution ratio is 100 percent, indicating that the excess PEA^+^ causes the QD to disintegrate. The main reasons for the performance improvement are as follows: (1) PEA^+^ is a two-dimensional material. Compared to 3D materials, PEA^+^ has stronger dielectric and quantum confinement effects, which allows it to possess stronger exciton binding energies, withstand greater thermal interference, and be more stable to light and humidity, leading to a considerable improvement in luminescence intensity and PLQY. As the dimensionality decreases, the crystal particle size decreases, leading to a stronger quantum confinement effect, which increases the band gap and blueshift of the PL and absorption spectra, as shown in Figure 7c. (2) PEA^+^ can increase the protonation degree of OAm ligand, thus promoting the binding of the surface of quantum dots with Br^−^, thereby reducing the density of defective states and the path of non-convergent radiation [25]. (3) The partial substitution of the high-insulated ligand OAm with the short-chain ligand PEA^+^ can enhance carrier injection and transport. (4) As mentioned in Section 3.2, the introduction of Br^−^ makes the halogen-abundant surface of the quantum dots passivate surface defects and enhance radiation convergence. At the same time, the formation of PbBr_n_ compounds can also effectively act as a shell for QDs, improving stability.

After partially substituting ligand OAm with PEABr in quantum dots, as depicted in Figure 7b, we observed that the luminescence intensity remained stable for 3 days and the luminescence wavelength only shifted from 516 nm to 518 nm. Additionally, we noted a significant improvement in the clustering phenomenon compared to the control group. This suggests that the substitution process has a positive effect on the stability of quantum dots, which is due to the fact that the substituted ligands have stronger exciton binding energies, lower defect density of states, and enhanced carrier injection and transport capabilities. As a result, this improves the optical properties and stability of the QD. From the XRD pattern of Figure 7d, it can be seen that the diffraction peaks at 26.5° and 37.8° correspond to the (111) and (211) planes of perovskite, indicating that the crystal structure of the quantum dot after substituting the ligand has no clear shift.

### 3.4. CsPbBr_3_ PQDs with PEABr and OAm Ligand Exchanges

The ligand PEABr was used in this study to exchange ligands with the produced QDs in percentages of 25%, 50%, 75%, and 100%. The experimental findings are displayed in Figure 8 and Table 1. The ligand-exchanged CsPbBr_3_ PQDs displayed a vivid blue-green hue under UV light and a clear light yellow color in the visible spectrum. The solution of QDs after ligand exchange is clearer when compared to those in Groups I and III, and the luminescence color is shifted to short-wavelength blue, indicating that the size of QDs is reduced and the luminescence performance is improved after ligand exchange via PEABr. However, the effect is not as clear as that of direct ligand substitution before synthesizing QDs in Group III, which is why Group III uses this method. The aggregation degree and luminescence attenuation intensity are lower than those in Group I but higher than those in Group III, which indicates that the stability and luminescence of the quantum dots after ligand substitution have been improved to a certifiable degree. As time passes, the solution after ligand exchange becomes turbid and the fluorescence intensity is weakened, which indicates that the stability of QDs decreases, leading to the enlargement and agglomeration of QDs.

FRET can happen between any fluorophores that show spectral overlap between the donor emission and acceptor excitation, but how it behaves depends on the population of fluorophores that are present. There is a significant amount of spectral overlap between the luminescence spectra of slightly smaller donor QDs and the absorption spectra of slightly larger acceptor QDs in the nominally monochromatic population where homo-transfer, or energy transfer between multiple copies of the same molecule, occurs in quantum dots (QDs) [26,27,28]. Thus, homo-transfer is characterized by a quenching of the blue luminescence and an amplification of the red luminescence inside the monochromatic peak, resulting in a redshift of the emission peak and a narrowing of the emission linewidth (Figure 9a) [29]. In this case, PLQY increased to 76.4% from 60.2% before the exchange. However, in contrast to the ligand substitution in Section 3.3, the intensity of the lift-off is significantly reduced, and the PL pattern only slightly changes or even slightly decreases as the ligand exchange fraction increases. This is due to the fact that ligand substitution after QD synthesis is more difficult than ligand substitution directly before synthesis, and the amount of OAm ligands that can be substituted is limited. Although no matter how often PEABr is added, it has reached a saturation amount capable of exchange, and abundant ligand and cation ions may also conduct rapid ionic exchange on the QD surface, reducing the stability and optical properties of the QD. As a result, there is only a limited amount of PL strength and blueshift that can be improved, and there is also a limited amount of stability improvement. As shown in Figure 9b, the intensity of the luminescence does not decrease after 3 days, but the wavelength position of the luminescence is shifted from 526 nm to 531 nm. Compared to the control group, the agglomeration phenomenon is somewhat improved, but there is still a certain gap compared to ligand substitution. It can also be seen from the absorption map in Figure 9c that the absorption of the QD after ligand exchange is significantly higher than that of the control group, but not as strong as the lifting strength of ligand substitution, in agreement with previous results. From the XRD pattern of Figure 9d, it can be seen that the diffraction peaks at 26.5° and 37.8° correspond to the (111) and (211) crystal planes of perovskite, indicating that the crystal structure of the quantum dot after the exchange of ligands does not modify significantly.

### 3.5. DA Replaces the CsPbBr_3_ PQDs of the Ligand

In this study, the long-chain oleic acid was totally replaced with the short-chain ligand DA, and the experimental outcomes are displayed in Figure 10 and Table 1. In comparison to Group I, the solution of CsPbBr_3_ PQDs after replacing the DA ligand is clearer and the luminescence color is shifted to a shorter wavelength blue color, indicating that the size of the quantum dots after replacing the ligand is reduced. The initial solution of the CsPbBr_3_ PQDs exhibits a clear green color under daylight and UV light, and a clear green color under UV light. The performance of the luminescence was enhanced.

To explore the effect of DA on the optical properties of QDs after and before the replacement of ligand OA by DA, we test the comparison between the ligand OA after and before the replacement by DA. As shown in Figure 11a, after the ligand OA is replaced by DA, the PL emission intensity increases to a certain extent and the emission wavelength is blue-shifted. As shown in Table 2, PLQY increased from 60.2% to 83.5% before substitution. The compelling reason for this lies in the potent attachment between the short-chain ligand DA and the surface of the quantum dot. This attachment significantly diminishes the loss of surface ligands, ultimately resulting in superior surface passivation, minimal surface defects, and reduced non-radiative recombination [30]. At the same time, the conductivity of the short-chain ligand, which replaces the original long-range insulating ligand, is significantly improved and the carrier transport rate is accelerated, which is favorable for luminescence. As shown in Figure 11b, the intensity of the luminescence does not decrease after 3 days, the wavelength of the luminescence is only redshifted from 516 nm to 519 nm, and the stability is slightly improved compared to the control. It can also be seen from the absorption map in Figure 11c that the QD absorption after ligand substitution is significantly higher than that of the control group, indicating an improvement in its optical performance, in agreement with previous results. From the XRD pattern of Figure 11d, it can be seen that the diffraction peaks at 15.2°, 26.5°, 30.6°, and 37.8° correspond to (100), (111), (200), and (211) crystal planes of perovskite, and there are two additional CsPbBr_3_ characteristic peaks than before ligand substitution, but it is still the CsPbBr_3_ PQD structure.

### 3.6. PMMA Encapsulated CsPbBr_3_ PQDs

The surface coating approach efficiently increases the QDs resistance to water and oxygen by greatly reducing the deterioration of the QDs, which is accomplished primarily by the use of transparent wide bandgap materials to encapsulate and shield the QDs from the external environment. In addition, it prevents the loss of quantum efficiency brought on by aggregation and ion exchange effects. The quantum dots underwent encapsulation using the widely-used polymer, PMMA, in this research. Figure 12 and Table 1 showcase the experimental outcomes. The PMMA-encased CsPbBr_3_ PQDs solution displays transparency and a green hue in natural light, while emitting bright light when exposed to ultraviolet light. The encapsulated QDs solution is significantly cleaner and more transparent than Group I, which indicates an improved luminescence performance. The sunlight-exposed solution tends to become slightly turbid with time, but even after 8 months, the solvent does not completely evaporate, leaving behind a relatively clear, light-yellow solution that still fluoresces emerald green when exposed to UV light (Figure 12f). This highlights a noticeable increase in the solution’s stability.

To explore the impact of different ratios of PMMA encapsulation on the stability and optical properties of QDs, we tested QDs with PMMA encapsulation QD: PMMA = 1:2, QD: PMMA = 1:1, QD: PMMA = 2:1, and compared them to those without PMMA, as shown in Figure 13a. After packing with different ratios of PMMA, the PL emission intensity increases significantly and the emission wavelength is blue-shifted. The intensity of PL emission is at its highest when the QD: PMMA packing ratio is 2:1, as shown in Table 2. This indicates that the size of QDs is reduced and the luminosity performance is improved. PLQY increases from 60.2% to 90.1%, which is the highest improvement among the optimization methods. PMMA acts as an ideal protective layer and isolates external factors like water and oxygen that can affect the stability of QDs. Based on the data presented in Figure 13b, it can be observed that the luminosity remains constant even after 3 days. Additionally, there is only a slight shift in the luminosity wavelength from 520 nm to 521 nm, indicating an impressive level of stability when compared to the control group. It can also be seen from the absorption diagram in Figure 13c that the absorption of packaged quantum dots is significantly higher than that of the control group, indicating that the improvement of its stability is also conducive to the improvement of optical performance, which is consistent with the previous results. Upon analysis of the XRD pattern for Figure 13d, it is evident that the peaks detected at 15.2°, 26.5°, 30.6°, and 37.8° correspond to the (100), (111), (200), and (211) crystal planes of perovskite, respectively. Additionally, there are two distinct peaks characteristic of CsPbBr_3_ when compared to the previous packaging, yet it remains consistent with the CsPbBr_3_ PQD structure.

To measure the size of the CsPbBr_3_ PQDs, the QDs packed with the relatively stable PMMA were chosen for the TEM and HRTEM tests in this paper to minimize the effect of the clustering and size increase due to instabilities in the measurement process. As shown in Figure 14a,b, we can see that the size of the QD is between 10 nm and 15 nm, and the lattice spacing is 0.34 nm. As shown in Figure 14c, the average size of the QD is about 8.7 nm.

The smaller the size, the more pronounced the effect of size-dependent quantum confinement. The smaller size also makes it more susceptible to surface effects due to the larger specific surface area. It has been shown that quantum confinement and surface effects greatly impact the stability and luminescence properties of QDs, in agreement with the above description.

## 4. Conclusions

To summarize, we have made significant advancements in improving the stability and optical properties of QDs through a variety of effective methods. These methods involve utilizing varying quantities of MnBr_2_ for doping, replacing or exchanging ligands such as OAm with different amounts of PEABr, exchanging OA with DA, and encapsulating CsPbBr_3_ PQDs with different levels of PMMA. Notably, our PMMA encapsulation technique has yielded the highest PLQY for QDs, elevating it from 60.2% to an impressive 90.1%, while simultaneously providing superior stability.

## Figures and Tables

**Figure 1 nanomaterials-13-02372-f001:**
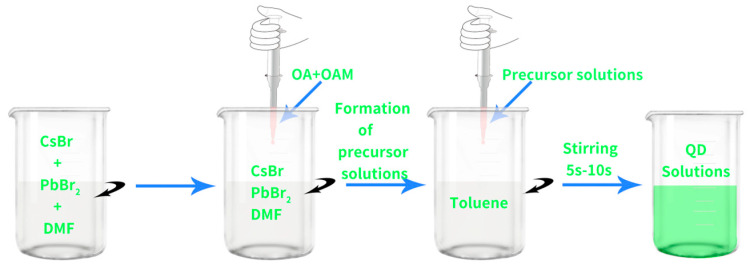
Flowchart of CsPbBr_3_ PQDs preparation.

**Figure 2 nanomaterials-13-02372-f002:**
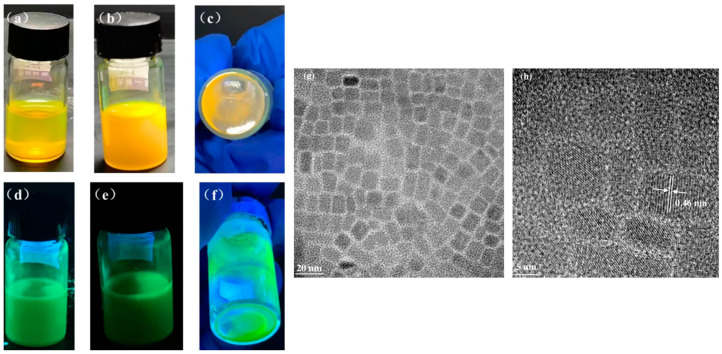
Photos of the original CsPbBr_3_ PQDs in solution under sunlight (**a**) day 0, (**b**) after 1 month, (**c**) after 8 months, and under UV light (**d**) day 0, (**e**) after 1 month, and (**f**) after 8 months, (**g**) CsPbBr_3_ PQDs TEM spectra and (**h**) HRTEM spectra.

**Figure 3 nanomaterials-13-02372-f003:**
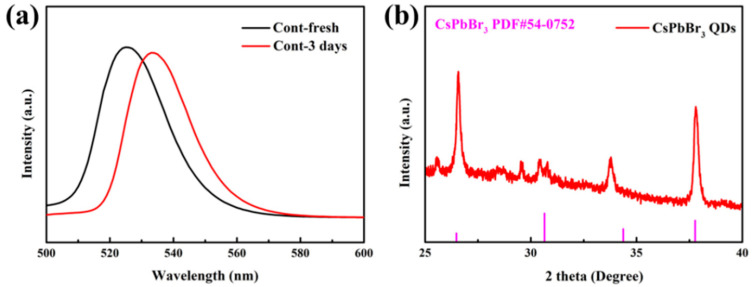
(**a**) PL and (**b**) XRD patterns of the original CsPbBr_3_ PQDs at initial and 3 days later.

**Figure 4 nanomaterials-13-02372-f004:**
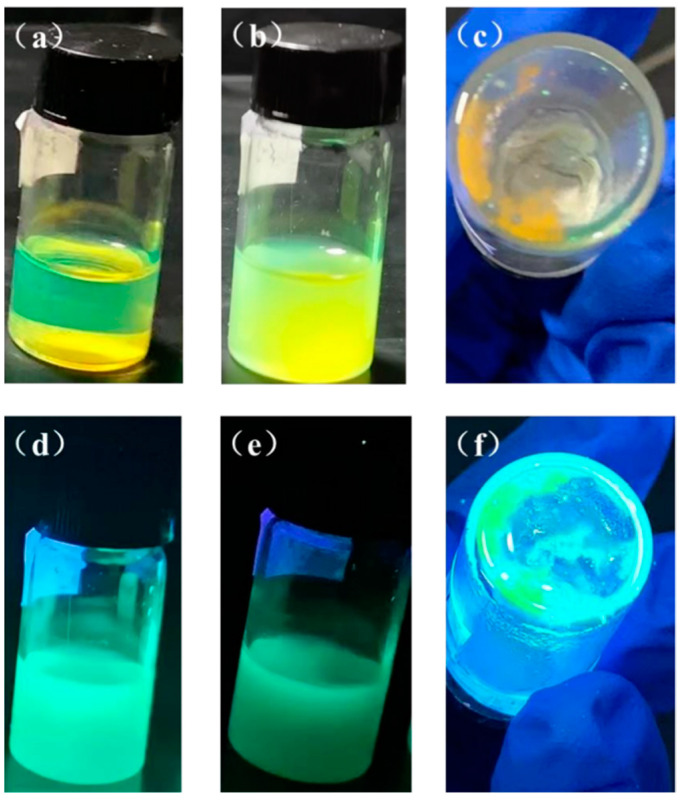
Solution photos of Mn^2+^ doped CsPbBr_3_ PQDs under sunlight (**a**) day 0, (**b**) after 1 month, (**c**) after 8 months, and under ultraviolet light (**d**) day 0, (**e**) after 1 month, and (**f**) after 8 months.

**Figure 5 nanomaterials-13-02372-f005:**
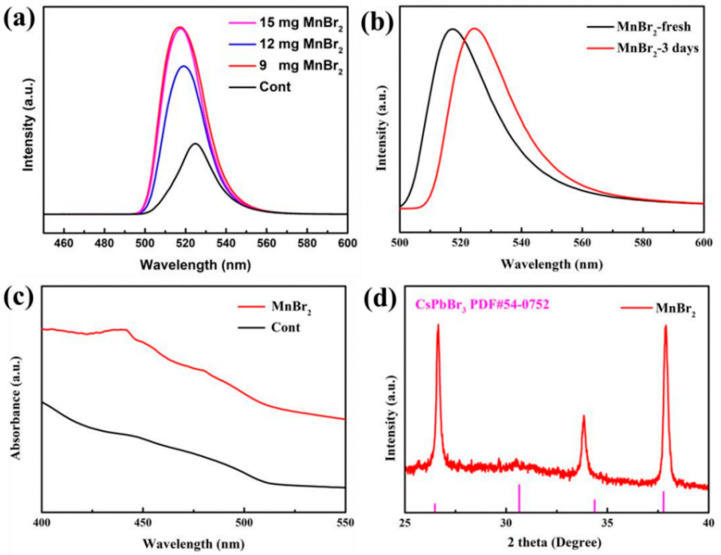
(**a**) CsPbBr_3_ PQDs PL spectra of Mn^2+^ doping with different contents, (**b**) PL spectra of Mn^2+^ doping CsPbBr_3_ PQDs at initial and after 3 days, (**c**) absorption spectra, (**d**) XRD spectra.

**Figure 6 nanomaterials-13-02372-f006:**
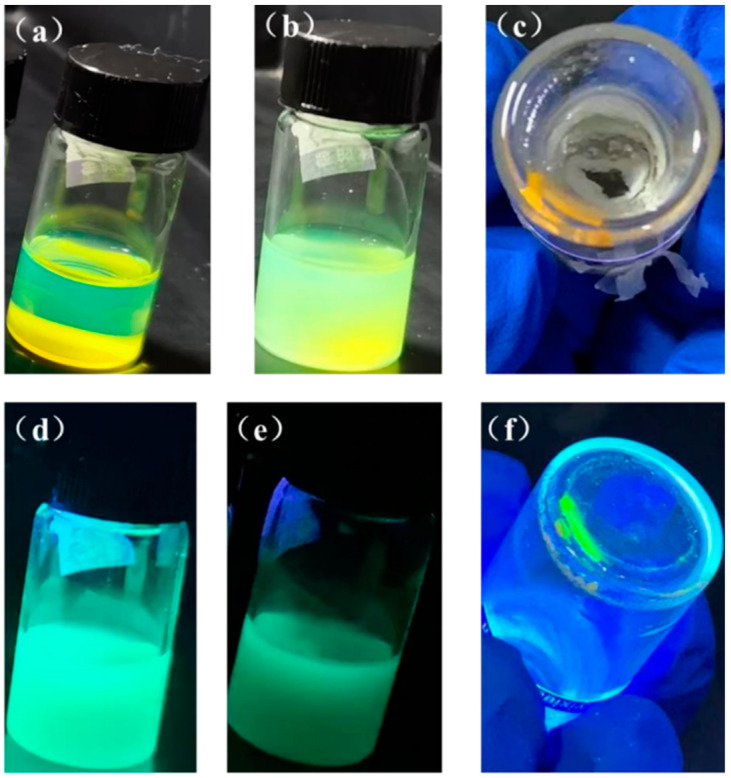
Solution photos of PEABr partially substituted ligand CsPbBr_3_ PQDs under sunlight (**a**) day 0, (**b**) after 1 month, (**c**) after 8 months, and under ultraviolet light (**d**) day 0, (**e**) after 1 month, and (**f**) after 8 months.

**Figure 7 nanomaterials-13-02372-f007:**
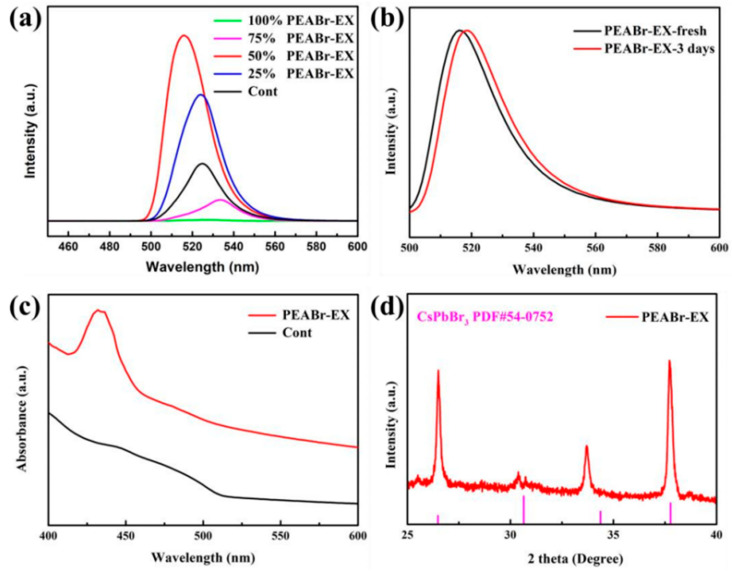
(**a**) CsPbBr_3_ PQDs PL spectra of PEABr partially substituted ligand OAm in different proportions, (**b**) PL spectra at initial and after 3 days, (**c**) absorption spectra, and (**d**) XRD spectra.

**Figure 8 nanomaterials-13-02372-f008:**
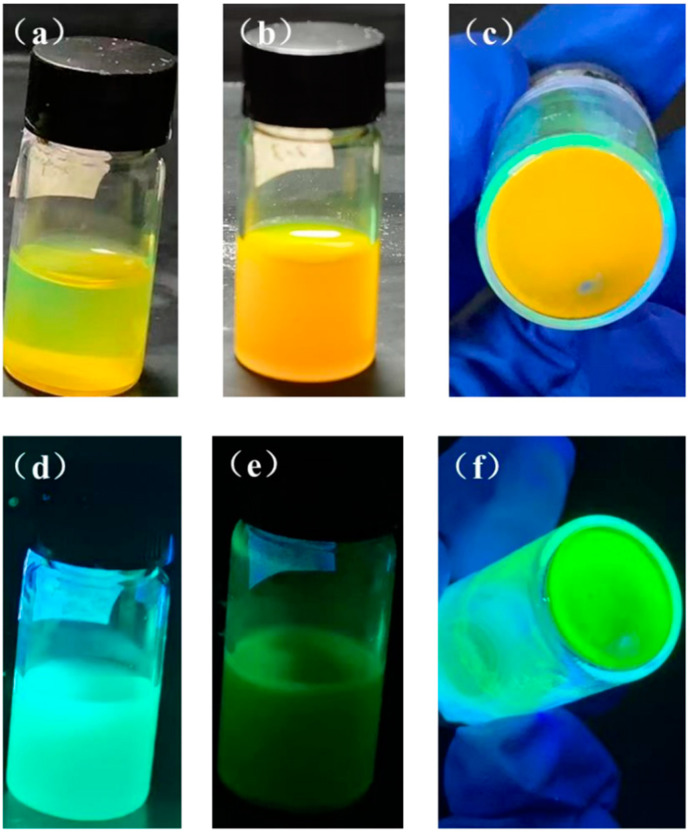
Solution photos of CsPbBr_3_ PQDs of PEABr partial exchange ligand (**a**) day 0 under sunlight, (**b**) after 1 month, (**c**) after 8 months, and (**d**) day 0, (**e**) after 1 month, and (**f**) after 8 months under UV light.

**Figure 9 nanomaterials-13-02372-f009:**
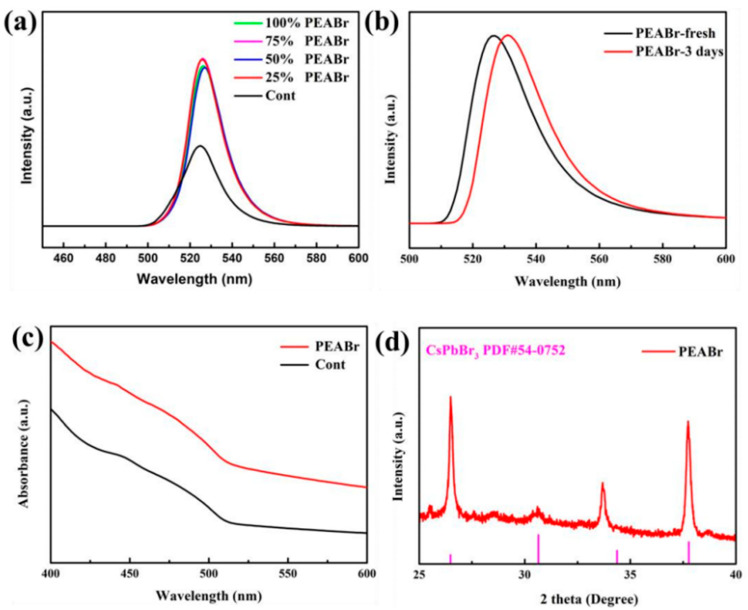
(**a**) CsPbBr3 PQDs PL spectra of the PEABr partial exchange ligand OAm in different proportions, (**b**) PL spectra at initial and after 3 days, (**c**) absorption spectra, and (**d**) XRD spectra.

**Figure 10 nanomaterials-13-02372-f010:**
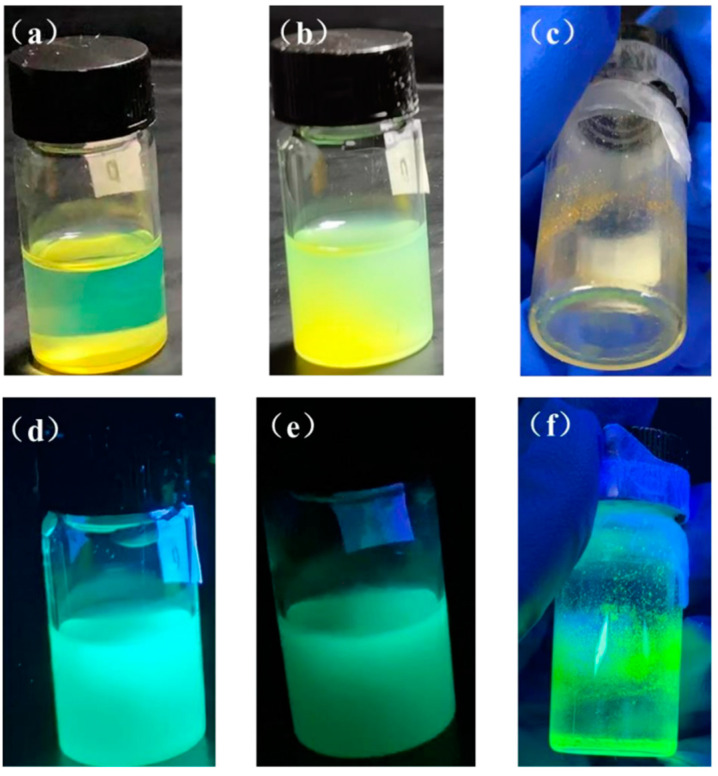
Solution photos of DA substituted ligand CsPbBr_3_ PQDs under sunlight (**a**) day 0, (**b**) after 1 month, (**c**) after 8 months, and under ultraviolet light (**d**) day 0, (**e**) after 1 month, (**f**) after 8 months.

**Figure 11 nanomaterials-13-02372-f011:**
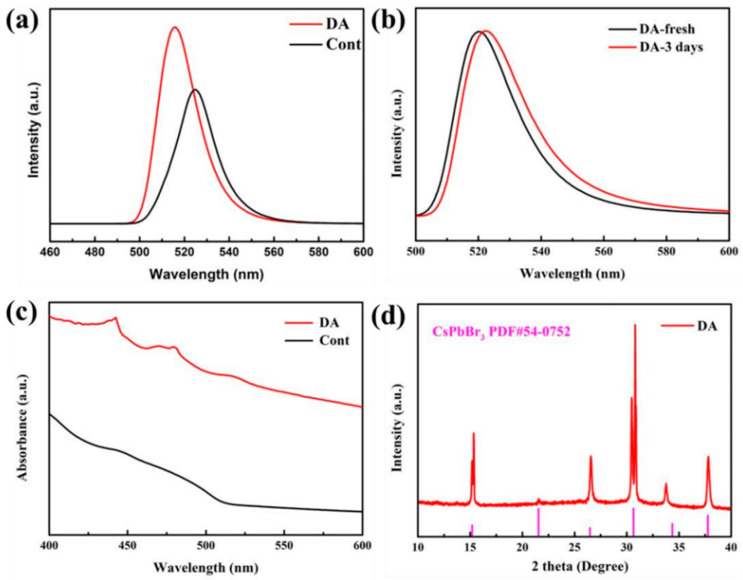
(**a**) CsPbBr_3_ PQDs PL pattern of DA replacement ligand OA, (**b**) PL pattern at initial and after 3 days, (**c**) absorption pattern, (**d**) XRD pattern.

**Figure 12 nanomaterials-13-02372-f012:**
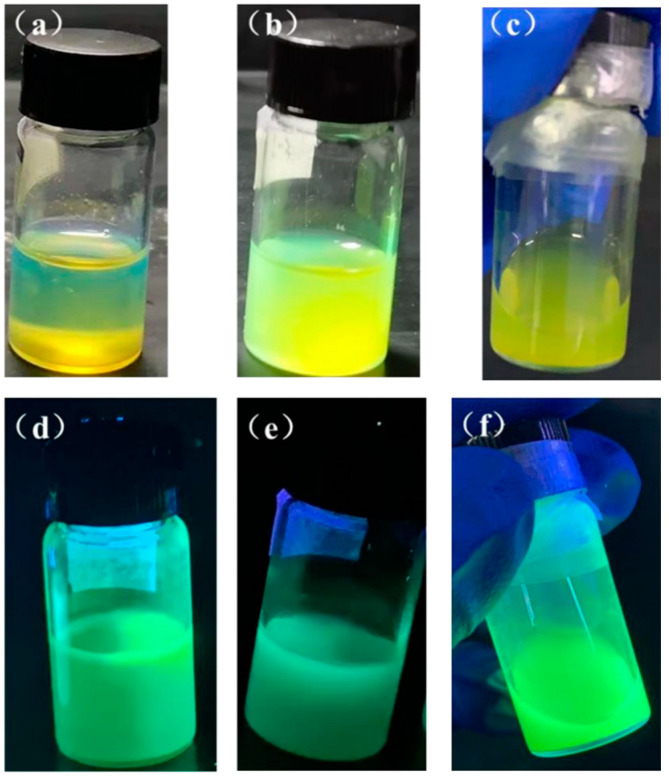
Solution photos of CsPbBr_3_ PQDs encapsulated by PMMA under sunlight (**a**) day 0, (**b**) after 1 month, (**c**) after 8 months and under UV light (**d**) day 0, (**e**) after 1 month, (**f**) after 8 months.

**Figure 13 nanomaterials-13-02372-f013:**
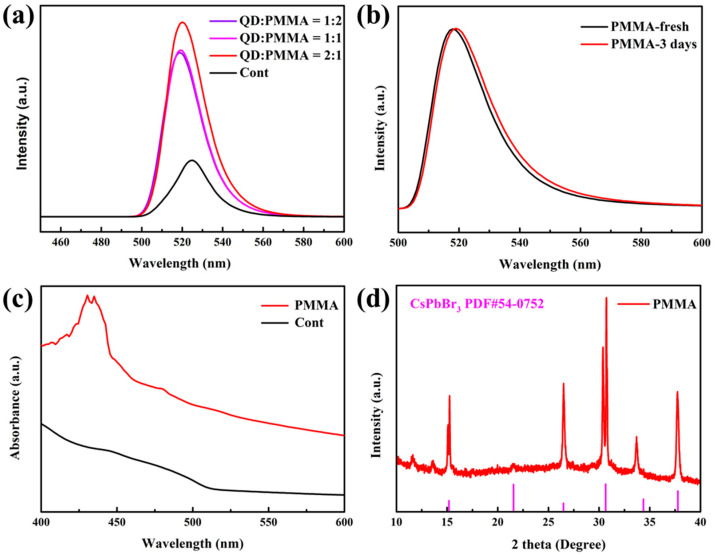
(**a**) PMMA encapsulated CsPbBr_3_ PQDs PL spectra, (**b**) PL spectra at initial and after 3 days, (**c**) absorption spectra, and (**d**) XRD spectra.

**Figure 14 nanomaterials-13-02372-f014:**
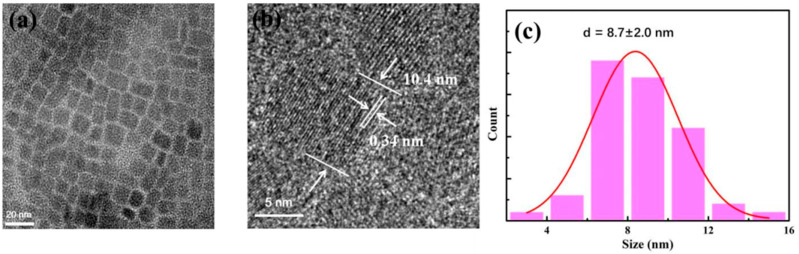
(**a**) PMMA encapsulated CsPbBr_3_ PQDs TEM spectra, (**b**) HRTEM spectra, and (**c**) size statistics spectra.

**Table 1 nanomaterials-13-02372-t001:** Summary of fluorescence intensity of CsPbBr_3_ PQDs initially, after 1 month, and 8 months.

Group	Name	Initial PL Emission Intensity (a.u.)	PL Intensity (a.u.) after 1 Month	PL Intensity (a.u.) after 8 Months
I	Original CsPbBr_3_ PQDs	1.7 × 10^6^	6.7 × 10^5^	1.5 × 10^4^
II	Mn^2+^ doped CsPbBr_3_ PQDs	2.0 × 10^6^	7.3 × 10^5^	1.6 × 10^4^
III	PEABr partially replaces the CsPbBr_3_ PQDs of the ligand	2.1 × 10^6^	9 × 10^5^	2.1 × 10^4^
IV	PEABr part allocation body exchange CsPbBr_3_ PQDs	1.9 × 10^6^	6.9 × 10^5^	2.0 × 10^4^
V	DA replaces the CsPbBr_3_ PQDs of the ligand	1.8 × 10^6^	6.7 × 10^5^	1.6 × 10^4^
VI	PMMA encapsulated CsPbBr_3_ PQDs	2.6 × 10^6^	1.3 × 10^6^	3.3 × 10^5^

**Table 2 nanomaterials-13-02372-t002:** Summary of emission wavelengths of CsPbBr_3_ PQDs initially, after 1 month, and 8 months.

Group	Name	Initial PL Emission Wavelength (nm)	After 3 Days PL Emits Wavelength (nm)	Fluorescence Quantum Yield (%)
I	Original CsPbBr_3_ PQDs	525	534	60.2
II	Mn^2+^ doped CsPbBr_3_ PQDs	517	525	78.0
III	PEABr partially replaces the CsPbBr_3_ PQDs of the ligand	516	518	85.2
IV	PEABr part allocation body exchange CsPbBr_3_ PQDs	526	531	76.4
V	DA replaces the CsPbBr_3_ PQDs of the ligand	516	519	83.5
VI	PMMA encapsulated CsPbBr_3_ PQDs	520	521	90.1

## Data Availability

The systematic article data used to support the findings of this study are included in the article.

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
