# Peer review of "Improvement of the Stability and Optical Properties of CsPbBr3 QDs"

_nanomaterials, 2023, doi:10.3390/nano13162372_

Round 1

Reviewer 1 Report

1. What is the novelty of the work? The described approaches as well as materials used are common. The authors need to highlight novelties.

2. In lines from 138 through 166, retelling well-known concepts is redundant. Moreover, there are mistakes there. For example,

Line 139: should be recombination instead of combination;

Line 140: “Excitons are produced by photoluminescence (PL)…” is an incorrect statement.

  1. In paragraphs from 3.1 through 3.6, a verbal description of the change in color of the solution over time lengthens the text. Description of experimental data should be presented in the form of tables, absorption and transmission spectra of colloid QDs solutions.
  2. In paragraph 3.1, the size of the original CsPbBr3 PQDs should be demonstrated (TEM and HRTEM).
  3. The transfer of excitation energy between PQDs upon changing ligands is not considered.

Author Response

  1. What is the novelty of the work? The described approaches as well as materials used are common. The authors need to highlight novelties.
  • Thank you very much for your The novelty of the work has been highlighted in Introduction Partas follows:
  • Page 2, the 2ndparagraph: In order to improve the stability of CsPbBr3 PQDs, five methods were adopted to optimize the quantum dots: (1) different amounts of Mn2+ doping; (2) different proportions of PEABr partially substituted ligand OAm before QDs synthesis; (3) different proportions of PEABr and OAm ligand exchange after QDs synthesis; (4) DA substituted ligand OA; (5) different proportions of PMMA encapsulation QDs. This paper focuses on the stability and fluorescence enhancement strategies of all-inorganic CsPbBr3 QDs, primarily describing the aspects of doping engineering, surface ligand modification, and surface coating strategies.”

  1. In lines from 138 through 166, retelling well-known concepts is redundant. Moreover, there are mistakes there. For example,

Line 139: should be recombination instead of combination;

Line 140: “Excitons are produced by photoluminescence (PL)…” is an incorrect statement.

  • Thank you very much for your These well-knownconcepts have been deleted according to your comments.

  1. In paragraphs from 3.1 through 3.6, a verbal description ofthe change in color of the solution over time lengthens the text. Description of experimental data should be presented in the form of tables, absorption and transmission spectra of colloid QDs solutions.
  • Thank you very much for your Description ofthe change in color of the solution over time lengthens has be modified as follows:
  • Page 3, the 7thparagraph: “We examined the fluorescence intensity of the QD solution after 1 and 8 months respectively, and compared it with the original solution in order to investigate the stability of CsPbBr3 PQDs in air, as shown in Fig. 2 and Table 1. According to the experimental findings, CsPbBr3 PQDs grow and cluster more with time while their fluorescence intensity diminishes because of their inherent instability.”

Table 1. Summary of fluorescence intensity of CsPbBr3 PQDs initially, after one month and eight months.

Group

Name

Initial PL emission intensity (a.u.)

PL intensity (a.u.) After 1 month 

PL intensity (a.u.) After 8 months

I

Original CsPbBr3 PQDs

1.7×106

6.7×105

1.5×104

II

 Mn2+ doped CsPbBr3 PQDs

2.0×106

7.3×105

1.6×104

III

PEABr partially replaces the CsPbBr3 PQDs of the ligand

2.1×106

9×105

2.1×104

IV

PEABr part allocation body exchange CsPbBr3 PQDs

1.9×106

6.9×105

2.0×104

V

DA replaces the CsPbBr3 PQDs of the ligand

1.8×106

6.7×105

1.6×104

VI

PMMA encapsulated CsPbBr3 PQDs

2.6×106

1.3×106

3.3×105

  1. In paragraph 3.1, the size of the original CsPbBr3PQDs should be demonstrated (TEM and HRTEM).
  • Thank you very much for your In paragraph 3.1, the size of the original CsPbBr3PQDs has been demonstrated as follows.
  • Page 3, the 1stparagraph: “ Besides, from Figs. 2g and 2h, we can obtained that the QDs have a size of around 10 nm and a lattice spacing of 0.46 nm.” 
  1. The transfer of excitation energy between PQDs upon changing ligands is not considered.
  • Thank you very much for yourquestion. A brief description about the transfer of excitation energy between PQDs upon changing ligands has been added as follows:
  • Page 10, the 1stparagraph: “FRET can happen between any fluorophores that show spectral overlap between the donor emission and acceptor excitation, but how it behaves depends on the population of fluorophores that are present. There is a significant amount of spectral overlap between the luminescence spectra of slightly smaller donor QDs and the absorption spectra of slightly larger acceptor QDs in the nominally monochromatic population where homo-transfer, or energy transfer between multiple copies of the same molecule, occurs in quantum dots (QDs) [26-28]. Thus, homo-transfer is characterized by a quenching of the blue luminescence and an amplification of the red luminescence inside the monochromatic peak, resulting in a redshift of the emission peak and a narrowing of the emission linewidth (Figure 9a) [29].”

Reviewer 2 Report

The authors of the manuscript “Improve the stability and optical properties of CsPbBr3 QDs” described the improvement the stability of CsPbBr3 PQDs on the premise of ensuring their  excellent optical properties. The authors selected four materials to optimize the quantum dots by doping, changing ligand and polymer encapsulation: different amounts of  Mn2+ dopingdifferent proportions of PEABr partially substituted ligand OAm before QDs synthesis different proportions of PEABr and OAm ligand exchange after QDs  synthesisDA substituted ligand OAdifferent proportions of PMMA encapsulation QDs to improve its stability and optical properties and its mechanism of action were  analyzed. The test results show that the above five methods can improve the stability and luminescence performance of QDs, with the best stability achieved when PMMA encapsulates QDs with a ratio of PMMA = 2:1 and PLQY increases from 60.2% to 90.1%.

The manuscript is recommended for publication in the journal as presented.

 Minor editing of English language required

Author Response

The authors of the manuscript “Improve the stability and optical properties of CsPbBr3 QDs” described the improvement the stability of CsPbBr3 PQDs on the premise of ensuring their  excellent optical properties. The authors selected four materials to optimize the quantum dots by doping, changing ligand and polymer encapsulation: different amounts of  Mn2+ dopingdifferent proportions of PEABr partially substituted ligand OAm before QDs synthesis different proportions of PEABr and OAm ligand exchange after QDs  synthesisDA substituted ligand OAdifferent proportions of PMMA encapsulation QDs to improve its stability and optical properties and its mechanism of action were  analyzed. The test results show that the above five methods can improve the stability and luminescence performance of QDs, with the best stability achieved when PMMA encapsulates QDs with a ratio of PMMA = 2:1 and PLQY increases from 60.2% to 90.1%.

The manuscript is recommended for publication in the journal as presented.

Comments on the Quality of English Language: Minor editing of English language required.

  • Thank you very much for yourcomment The paper has been checked carefully, and modifications have been made to improve the English writing. 

Reviewer 3 Report

It must be improved. Major editing and consolidation are necessary. 

Author Response

The report presented by Wang et al. demonstrates several strategies for improving the stability of All inorganic perovskite QDs. The manuscript is interesting and represents progress towards pushing the optoelectronic properties of perovskites QDs. There are several concerns that the authors should address before I recommend this paper for publication. 

  1. The introduction should focus more on how the different synthetic approaches affect stability as well as optoelectronic properties.
  • Thank you very much for your The following descriptionhas been added to the introduction according to your comments:
  • Page 2, the 1stparagraph: “CsPbX3 QDs have low stability due to their high sensitivity to the outside environment, which limits the commercial use of perovskite[16]. To increase the stability of perovskite QDs, more research is still needed. As reported, the common routes for enhancing the stability of all-inorganic perovskite QDs include surface ligand modification, doping engineering, and surface coating. Zou et al. (2017) tried to substitute comparable ions for some of the elements at the A and B sites by ion-doping[17 ]. The A-site doping aids in raising the tolerance factor, and further enhances the stability of perovskite material. The B-site doping primarily substitutes Pb2+ with smaller ion radii, changing the length of the B-X bond in the process, which increases phase stability and reduces the number of dangerous Pb2+ Surface ligand modification aims to introduce ligands that firmly bind to the uncoordinated lead and halide ions on the QDs surfaces, passivating dangling bonds and defect sites, thereby reducing non-radiative channels and improving optical performance[18-20]. Liu et al. Reported a surface coating method, which adopted transparent wide-bandgap materials to encapsulate and shield the perovskite QDs from the external environment[21]. This sharply reduces the degradation of QDs, thereby improves their water and oxygen resistances. Additionally, it prevents the loss of quantum efficiency brought on by aggregation and ion exchange effects. After modification, the photoluminescence quantum yield (PLQY) and stability of perovskite QDs are both significantly increased.”

  1. The purpose of the paper is not well motivated in the introduction. Instead of what has beendone in this work, the authors should incorporate elements that inspired their synthetic 
  • Thank you very much for your The purpose of the paper and some relative elements have been motivated in the introduction as follows:
  • Page 2, the 2ndparagraph: In order to improve the stability of CsPbBr3 PQDs, five methods were adopted to optimize the quantum dots: (1) different amounts of Mn2+ doping; (2) different proportions of PEABr partially substituted ligand OAm before QDs synthesis; (3) different proportions of PEABr and OAm ligand exchange after QDs synthesis; (4) DA substituted ligand OA; (5) different proportions of PMMA encapsulation QDs. This paper focuses on the stability and fluorescence enhancement strategies of all-inorganic CsPbBr3 QDs, primarily describing the aspects of doping engineering, surface ligand modification, and surface coating strategies.”

  1. Page 4 section 3: “We understand that quantum dots glow through the combination of electrons and hole pairs (exciton annihilation)” ---this statement has significant flaws. Electron hole pair radiation recombination is the reason for glowing. Exciton annihilation is a completely different physical process and may not be related to radiative recombination. Exciton annihilation could be non-radiative. It is highly recommended that the authors use appropriate scientific language.
  • Thank you very much for yoursuggestions. Another referee pointed out that this part is redundant, so Page 4 section 3 has been delet

  1. Page 4 section 3: “Combine into excitons”-please correct this language. Band edge carriers do not form exciton as the authors described.
  • Thank you very much for yoursuggestions. Another referee pointed out that this part is redundant, so Page 4 section 3 has been delet

  1. Page 5, Line 183-189. Please explain the reason for red shifting and lower quantum yield in detail with proper literature citations.
  • Thank you very much for yourThe reason for red shifting and lower quantum yield have been explained as follows:
  • Page 4, the 2ndparagraph: “Fully inorganic halide-encapsulated crystalline quantum dots exhibit brightness loss because of their inherent instability under ambient circumstances. Additionally, because of their weak internal ionic bonding and high surface energy, these quantum dots agglomerate and experience structural deterioration. The quantum dots' capacity to absorb and emit energy declines, their surface energy rises, more of them are gathered together, and their emission spectra change toward lower energies, or toward becoming red[22].”

  1. The authors reported that doping with Mn2+reduces the size and increases the QY, however, appropriate reasoning for these observations is missing.
  • Thank you very much for your The reason forMn2+ reduces the size and increases the QY have been added as follows:
  • Page 5, the 3rdparagraph: “Part of the Pb2+ was replaced with Mn2+, which was effectively doped into the lattice of PQDs. The doping of Mn2+ resulted in the lattice contraction, which would decrease the size of quantum dots since Mn2+ ionic radius is less than Pb2+. Additionally, the amount of uncoordinated Pb2+ on the surface of the PQDs has been significantly reduced as a result of the doping of Mn2+, and Mn2+ has a higher affinity for the halogen group elements. These factors increase the formation energy, reduce the density of defect states on the surface, and improve the stability of PQDs, all of which contribute to a higher PLQY. ”

  1. The authors should include a description of why the ligand exchange and encapsulation increase the QY and reduce the non-radiative pathways.
  • Thank you very much for your The descriptionof why the ligand exchange and encapsulation increase the QY and reduce the non-radiative pathways have been added as follows:
  • Page 7, the 2ndparagraph: “In order to decrease non-radiative channels and improve optical characteristics, surface ligand modification intends to introduce ligand passivation of dangling bonds and defective sites of uncoordinated lead and halogen ions that are tightly attached to the QD surface.”
  • Page 12, the 1stparagraph: “The surface coating approach efficiently increases the QDs' resistance to water and oxygen by greatly reducing the deterioration of the QDs, which is accomplished primarily by the use of transparent wide bandgap materials to encapsulate and shield the QDs from the external environment. In addition, it prevents the loss of quantum efficiency brought on by aggregation and ion exchange effects.”

  1. The paper sounded very repetitive, can be consolidated and the writing can be improved a lot. It is recommended to improve the writing to help the readers understand the paper better.
  • Thank you very much for yourThe paper has been consolidated, and lots of modifications have been made to reduce the repetition rate and improve the writing. 

Round 2

Reviewer 1 Report

All responses to the comments are accepted. There remains a small remark: a title must be added to Table 2.

Author Response

  1. All responses to the comments are accepted. There remains a small remark: a title must be added to Table 2.
  • Thank you very much for your A title has been added to Table 2as follows: 
  • Table 2. Summary of emission wavelengths of CsPbBr3 PQDs initially, after one month, and eight months.

Reviewer 3 Report

The authors have improved the paper and addressed all the comments. 

The quality of English must be improved. Even in the response letter, there are several typos. I would highly recommend that the authors should read the manuscript carefully and improve the quality of the presentation. 

Author Response

  1. The quality of English must be improved. Even in the response letter, there are several typos. I would highly recommend that the authors should read the manuscript carefully and improve the quality of the presentation.
  • Thank you very much for yourI have read the manuscript carefully and improve the quality of the presentation, as highlighted in the text.